# Therapeutic Potential of Glutaminase Inhibition Targeting Metabolic Adaptations in Resistant Melanomas to Targeted Therapy

**DOI:** 10.3390/ijms26178241

**Published:** 2025-08-25

**Authors:** Laura Soumoy, Aline Genbauffe, Dorianne Sant’Angelo, Maude Everaert, Léa Mukeba-Harchies, Jean-Emmanuel Sarry, Anne-Emilie Declèves, Fabrice Journe

**Affiliations:** 1Laboratory of Human Anatomy & Experimental Oncology, Faculty of Medicine, Pharmacy and Biomedical Sciences, Research Institute for Health Sciences and Technology, University of Mons, 7000 Mons, Belgium; aline.genbauffe@umons.ac.be (A.G.); maude.everaert@alumni.umons.ac.be (M.E.); lea.mukeba-harchies@student.umons.ac.be (L.M.-H.); 2INSERM U981, Gustave Roussy Cancer Campus, 94800 Villejuif, France; 3Centre de Recherches en Cancérologie de Toulouse, Université de Toulouse, Inserm, CNRS, 31100 Toulouse, France; jean-emmanuel.sarry@inserm.fr; 4Laboratory of Metabolic and Molecular Biochemistry, Faculty of Medicine, Pharmacy and Biomedical Sciences, Research Institute for Health Sciences and Technology, University of Mons, 7000 Mons, Belgium; anne-emilie.decleves@umons.ac.be; 5Laboratory of Clinical and Experimental Oncology, Institut Jules Bordet, Université Libre de Bruxelles, 1000 Brussels, Belgium

**Keywords:** BRAF inhibitors, melanoma resistance, metabolic adaptations, glutaminase inhibition, CB-839, personalized therapy

## Abstract

Targeted therapy with BRAFi has significantly improved outcomes for patients with BRAF-mutated metastatic melanoma. However, resistance mechanisms, particularly metabolic adaptations, such as increased glutaminolysis, present substantial clinical challenges. This study investigated the metabolic changes underlying BRAFi resistance in melanoma cells. Using pharmacological agents, including dabrafenib (BRAFi), pimasertib (MEKi), dasatinib (cKITi), and CB-839 (glutaminase inhibitor), we explored metabolic adaptations in melanoma cell lines harboring various mutations. Our methodologies included cell culture, qPCR, polysome profiling, animal studies in nude mice, and analyses of patient samples to evaluate the therapeutic potential of targeting glutaminolysis. Our findings confirmed that melanoma cells, with resistance to targeted therapies, exhibit metabolic adaptations, including enhanced glutaminolysis, increased mitochondrial content, and elevated antioxidative capacities. We evaluated the efficacy of CB-839 and demonstrated its ability to reduce the proliferation of resistant melanoma cells both in vitro and in vivo. Mechanistic studies revealed that CB-839 suppressed ATP production and TCA cycle intermediates in resistant cells while inducing oxidative stress in sensitive cells, thereby inhibiting their proliferation. High glutaminase expression in primary patient tumor samples was associated with poor prognosis. We identified a metabolic signature in tumors from patients responsive or unresponsive to BRAFi prior to treatment, which could serve as a predictive factor for BRAFi response. This study underscores the metabolic alterations driving resistance to BRAFi in melanoma cells and highlights the therapeutic potential of targeting glutaminolysis with CB-839. The identification of metabolic signatures in patient samples provides valuable insights for personalized treatment strategies, aiming to overcome resistance mechanisms and improve patient outcomes in melanoma management.

## 1. Introduction

Melanoma is primarily driven by mutations in proteins of the MAPK pathway, such as BRAF, NRAS, NF1, or the RTK cKIT [1]. These mutations result in the constitutive activation of the MAPK pathway, significantly increasing the expression of genes involved in glycolysis and glucose transport, thereby rendering these cells highly glycolytic [2]. In this context, BRAF inhibitors (BRAFi) decrease glycolysis [3], and MEK inhibitors reduce lactate production in BRAF-mutated cells [4]. The antitumor effect of these inhibitors is, at least partially, linked to this blockade of glycolysis and the subsequent decrease in ATP production [5]. Nevertheless, resistance to these inhibitors rapidly develops [6]. We and others have highlighted a metabolic switch in cells developing resistance to BRAF or BRAF/MEK inhibitors towards the use of glutaminolysis to fuel the TCA cycle [7,8,9]. While these observations have been made in vitro, there is currently no evidence for a link between a metabolic switch towards glutaminolysis and a resistant phenotype to targeted therapy in patients. Glutamine was more recently demonstrated to be a key player in cancer [10,11,12]. Indeed, glutamine is the most abundant amino acid in human plasma [13], and it is crucial for many functions in cancer cells [14]. It can be converted into metabolites of the TCA cycle [15] and is also essential for the synthesis of glutathione [16], proteins, and nucleotides [17]. These metabolites are necessary for cancer cells to sustain their rapid proliferation rate. Glutamine is imported into the cells via the SLC1A5 transporter [18] and is then oxidized by glutaminase to form glutamate [15,19].

Glutaminase, the initial enzyme involved in glutaminolysis, is encoded by two genes: the kidney-type glutaminase (*GLS*) and the liver-type glutaminase (*GLS2*) [20]. GLS protein is expressed in various healthy tissues, while GLS2 is restricted to specific organs, such as the brain, liver, pancreas, and pituitary glands [21]. GLS is known to play a primary role in cancer cells [22]. Alternative splicing of *GLS* pre-mRNA leads to two isoforms: glutaminase C (GAC) and kidney-type glutaminase (KGA) [23], both of which are localized in mitochondria [24,25]. GAC and KGA are believed to be crucial in cancer development, as they are overexpressed in various types of cancers, including melanoma [26]. Furthermore, a study reported increased *GLS* gene expression in melanoma cells resistant to BRAFi [7].

CB-839 (Telaglenastat^®^) is an orally available, potent allosteric inhibitor of GAC and KGA [10,27,28]. It has already demonstrated potent antitumor effects in different cancer preclinical models, such as triple-negative breast cancer [29], colorectal cancer [30,31], glioblastoma, and nasopharyngeal carcinoma [32]. Interestingly, this molecule is already being evaluated in various clinical trials [33], but no studies have focused on the response of resistant melanoma cells to this inhibitor.

Therefore, in this study, we hypothesize that melanoma cells resistant to RTKi/MAPKi undergo a metabolic switch toward glutaminolysis, and that this dependency can be therapeutically targeted using the glutaminase inhibitor CB-839 to overcome resistance and restore antitumor efficacy. To demonstrate the hypothesis, we further characterized glutamine metabolism in melanoma cell lines (with mutations on BRAF, NRAS, or cKIT) that are resistant to RTKi/MAPKi and in tumor samples from melanoma patients. Also, we evaluated the impact of CB-839 on decreasing melanoma resistance to single targeted therapy, both in vitro and in vivo (BRAF mouse model).

## 2. Results

### 2.1. Melanoma Cells with Acquired Resistance Depend on Glutamine and Rely on Glutaminolysis for Their Survival

First, glutamine and glutamate levels were measured in sensitive cells, both untreated and after 24 h of exposure to RTKi/MAPKi, as well as in resistant cells. As illustrated in Figure 1A, the glutamine/glutamate ratio was significantly decreased in the three cell lines treated with inhibitors, as well as in the three resistant cell lines compared to their sensitive counterparts, suggesting an increased conversion of glutamine into glutamate, especially in the MM074 line.

We then investigated the effect of glutamine deprivation on apoptotic cell death. Our results demonstrated a significant increase in apoptosis induction in resistant cells cultured for 48 h in a glutamine-free medium, whereas no apoptotic induction was observed in sensitive cells under the same conditions (Figure 1B). Additionally, we conducted a clonogenic assay on cells cultured with 0, 0.5, or 1 mM of glutamine. A non-significant reduction in colony formation was observed at 0.5 mM, while a substantial and statistically significant decrease was detected at 0 mM in resistant cells. In contrast, sensitive cells showed no significant changes in colony formation under these conditions (Figure 1C). This finding further supports the dependency of resistant cell lines on glutamine.

We then quantified the gene expression of key enzymes involved in glutaminolysis (*GAC*, *KGA*, *GLS2*, *GSH*) and glutamine transporters (*SLC1A5*, *SLC7A5*, *SLC38A2*) through qPCR analyses. Our results indicated a significant upregulation of mRNA expression for the *GAC* and *GLUD1* enzymes, as well as for the SLC1A5 transporter in resistant cells compared to sensitive ones (Figure 1D). Furthermore, we conducted polysome profiling to validate the qPCR results on the limiting enzymes of glutaminolysis, *GAC*, and *GLUD1*. We observed an increased level of *GAC* and *GLUD1* mRNA in the heavy polysome fractions of resistant MM074-R cells compared to sensitive MM074 cells, indicating enhanced translation of these mRNAs in the resistant cell line (Figure 1E). Notably, these changes were less pronounced in the NRAS-mutated MM161 and MM161-R cells. This difference could be attributed to the fact that these cells, in their sensitive state, already exhibit higher levels of glutaminolysis compared to the HBL cell line and the highly glycolytic MM074 cell line (Appendix A).

### 2.2. Resistant Cells Have Higher Mitochondrial Content and Better Antioxidative Defenses

As glutaminolysis occurs in the mitochondria, we performed flow cytometry using MitoTracker Green to assess mitochondrial content in the cells. MitoTracker Green is a green fluorescent dye that passively diffuses across the plasma membrane and accumulates in the mitochondria of live cells. Our data showed that MM074-R and HBL-R resistant cells exhibited significantly higher mitochondrial content compared to their sensitive counterparts (Figure 2A). No change was observed in the MM161 line between sensitive and resistant cells. Notably, treatment of sensitive cells with RTKi/MAPKi significantly decreased mitochondrial levels in all cases.

Subsequently, we investigated the gene expression of co-activators and transcription factors involved in mitochondrial biogenesis. A significant upregulation of *PPARGC1A* and *TFAM* expression was observed in resistant cells (Figure 2B).

As glutaminolysis and mitochondrial metabolism generate higher amounts of ROS than glycolysis, we conducted flow cytometry to study the levels of total ROS (H2DCFDA) and mitochondrial ROS (Mitosox). These results showed significantly increased levels of both total and mitochondrial ROS in sensitive MM074 cells treated with dabrafenib, only significantly increased levels of mitochondrial ROS in sensitive HBL cells treated with dasatinib, whereas no changes were observed in the MM161 line. The levels of total ROS were also weakly increased in resistant MM074-R cells cultured with dabrafenib, while the levels of mitochondrial ROS were similar in resistant and untreated sensitive cells (Figure 2C). These findings suggest that resistant cells are more efficient at counteracting RTKi/MAPKi-induced ROS, likely due to an enhanced capacity to manage ROS accumulation under chronic treatment. To confirm this, we investigated the gene expression of the master regulator of antioxidant defense, the transcription factor *NRF2*, and its target genes. We observed significant increases in gene expression of *NRF2*, as well as glutathione reductase and peroxidase, in MM074-R and MM161-R cells. Catalase expression was also increased in MM074-R cells compared to their sensitive counterparts. Notably, no significant changes were detected between HBL-R and HBL cells (Figure 2D).

Finally, we conducted immunofluorescence analysis to confirm the activity of NRF2 in resistant cells. As illustrated in Figure 2E, and quantified in Appendix A, significant increases in nuclear NRF2 levels were observed in all three resistant cell lines compared to the sensitive ones. Moreover, even though none of its target genes were upregulated, the nuclear localization of NRF2 in HBL-R cells could suggest its activity.

### 2.3. Glutaminase Inhibitor CB-839 Blocks Glutaminolysis and Decreases Survival of Resistant Cells

To target the glutaminolysis pathway in resistant cells, we exposed them to the glutaminase inhibitor CB-839 and observed its impact on colony formation and cell apoptosis. The clonogenic assay indicated significant dose-dependent decreases in colony formation in resistant cells when treated with CB-839. The strongest effect was observed in the BRAF-mutated MM074-R cells, which exhibited the highest glutamine consumption (Figure 3A). We then studied apoptotic induction in resistant cells treated with CB-839 and observed a strong and significant induction of apoptosis in all resistant cell lines (Figure 3B).

Because the MM074-R cell line was the most sensitive to CB-839 (Figure 3A) and is clinically relevant in the context of BRAF inhibitor therapies, we specifically focus on both MM074 and MM074-R cells to validate whether CB-839 reduces cell survival by inhibiting glutaminolysis. To do so, we quantified glutamine and glutamate levels in MM074 and MM074-R cells and observed an increased glutamine/glutamate ratio in cells treated with CB-839, indicating a decrease in the conversion of glutamine into glutamate (Figure 3C). Next, we knocked down the expression of *GLS* using siRNA in MM074-R cells to confirm that this enzyme was the target of CB-839. As GLS is crucial for the metabolism of resistant cells, we observed an increased induction of apoptosis in KD cells (up to 30% apoptosis). However, in siRNA GLS cells, the induction of apoptosis by CB-839 was weaker than in scramble cells and was not significantly different from untreated cells in this case (Figure 3D). To further validate the specificity of CB-839, we performed a rescue experiment using α-ketoglutarate (α-KG). The addition of α-KG significantly reduced the antiproliferative effect of CB-839 and confirmed that the inhibition of cell proliferation was caused by a decrease in energy production induced by CB-839 treatment (Figure 3E).

Finally, to assess the involvement of oxidative stress in the effects of CB-839, we measured total glutathione (GSH) levels and observed a dramatic decrease in GSH in cells treated with CB-839 (Figure 3F), along with a significant increase in ROS levels (Figure 3G). However, supplementation with GSH did not reduce the antiproliferative effect of CB-839 (Figure 3H). This was observed despite a strong decrease in ROS levels (Figure 3G), suggesting that the antiproliferative effect of CB-839 is not associated with ROS production in resistant cells.

### 2.4. Glutaminolysis Inhibition Improves the Effectiveness of Targeted Therapy in Sensitive Cells by Increasing Oxidative Stress

We investigated the impact of the CB-839–RTKi/MAPKi combination in sensitive cells. We observed an increased effect of the combination on apoptosis induction (Figure 4A) and colony formation (Figure 4B) compared to either treatment alone, suggesting an enhanced efficacy when the two drugs are used together.

We then examined the effect of CB-839, alone or in combination with dabrafenib, in BRAF-mutated MM074 cells. Treatment with CB-839 led to a reduction in total GSH levels (Figure 4C). Additionally, we documented an increased level of ROS in cells treated with dabrafenib, CB-839, and even more with the combination. This ROS elevation was significantly reduced upon GSH supplementation (Figure 4D). Finally, we performed a rescue experiment with GSH to evaluate cell proliferation and observed a significantly reduced effect of CB-839 when GSH was added to the culture medium, compared to CB-839 alone (Figure 4E). Hence, in contrast to what was observed in resistant cells (Figure 3E,H), GSH but not α-KG enhanced the survival of sensitive cells exposed to CB-839 (Appendix A).

### 2.5. Glutaminolysis Inhibition Decreases Tumor Development in Mice Bearing BRAFi Resistant Xenografts

We investigated the antitumor effect of CB-839 in vivo in a mouse model xenografted with BRAF-mutated MM074-R cells (Figure 5A). One week after cell injection, mice were treated with CB-839 by oral gavage for 3 weeks. At the time of animal sacrifices, mice treated with CB-839 showed a significant reduction in both tumor volume and tumor weight compared to the control group (Figure 5B). Importantly, the treatment with CB-839 for 3 weeks did not induce animal toxicity, as demonstrated by the absence of significant animal weight changes in treated animals compared to the control ones (Appendix A).

### 2.6. The Glutaminolysis Enzyme GAC Is a Prognostic Marker in Metastatic Melanoma and a Signature Comprising GAC and Other Glutamine Metabolism-Related Enzymes May Predict Resistance to BRAFi

We evaluated the expression of GAC protein by immunohistochemistry (IHC) in 84 skin and lymph node metastases of melanoma patients (Table 1A). We observed varying staining intensities of GAC in melanoma tissues, which we scored on a scale from 0 to 3 (Figure 6A). This intensity score was then combined with the percentage of positively stained tissue to establish a score ranging from 0 to 100%. We searched for the optimal cut-off value by testing percentiles between 20 and 80, and by dividing the population at percentile 67 into low vs. high GAC groups, we demonstrated that high GAC protein expression was significantly associated with shorter patient overall survival (Figure 6B).

To confirm the metabolic switch observed in our cell lines and its impact on oxidative stress, we evaluated gene expressions in eight metastatic melanoma samples harboring the most frequent ^V600E^BRAF mutation and collected from patients prior to treatment with BRAFi (vemurafenib) (Table 1B). Among these patients, four were responders and four were non-responders to the targeted therapy. Regarding glutaminolysis enzymes (Figure 6C) and transporters (Figure 6D), we observed a significant increase in *GAC*, *KGA*, and *GLUD1* mRNA expression in unresponsive patients to the vemurafenib compared to responsive ones. Concerning mitochondrial biogenesis (Figure 6E) and antioxidative defenses (Figure 6F), we reported significant increases in *TFAM*, *PPRC1*, *NRF2* mRNA expression, and in the gene coding for glutathione reductase. Thus, a signature including all of these genes (*GAC*, *KGA*, *GLUD1*, *TFAM*, *PPRC1*, *NRF2*, *GSR1*) could be proposed as a predictive signature of response to BRAFi.

## 3. Discussion

We confirmed that melanoma cells with acquired resistance to targeted therapy exhibit more active glutamine metabolism and rely more heavily on glutaminolysis for survival and proliferation than their sensitive counterparts. This metabolic shift toward glutaminolysis was previously demonstrated in BRAF-mutant cells [7,8,34]. However, to our knowledge, this is the first study investigating the expression of genes involved in this shift across a panel of melanoma cell lines with different mutational profiles. Indeed, previous studies primarily focused on BRAF-mutant melanoma cells. Our study extends this analysis to a larger panel of melanoma cell lines, including non-BRAF mutants. This represents a critical advance in understanding glutaminolysis as a general resistance mechanism beyond the BRAF mutation context.

We also demonstrated that resistant cells have an increased mitochondrial pool associated with enhanced mitochondrial biogenesis, underscoring the crucial role of mitochondrial metabolism in these cells. Moreover, as mitochondrial metabolism is known to be associated with an increase in ROS production, we evaluated ROS levels and antioxidative defenses. Our results indicated that RTKi/MAPKi treatment induces oxidative stress and that, upon acquiring resistance, the cells develop stronger antioxidant capacities to survive in this more oxidative environment. While these observations regarding the mitochondrial pool, the amount of ROS, and the better antioxidative defenses of resistant cells were previously documented in the case of BRAF-mutated cells [35,36,37], they had not been characterized in other mutational backgrounds, as we have done here.

Of note, although the combination of dabrafenib (BRAFi) and trametinib (MEKi) is an established clinical treatment for BRAF-mutant melanoma due to its improved efficacy and delayed resistance, we employed single-agent treatments in our models to dissect their specific resistance mechanisms. This approach aligns with previous studies investigating resistance to targeted therapies [38,39] and is justified for several reasons. First, using a single agent allows for a more precise analysis of the molecular and metabolic changes driving resistance to BRAFi or MEKi individually. Second, because resistance often emerges in a stepwise manner, modeling resistance to one agent at a time reflects early adaptive events more clearly. Third, single-agent treatments facilitate comparability with extensive preclinical studies, where most resistance models were developed under BRAFi or MEKi monotherapy. Finally, differences in resistance mechanisms to BRAFi versus MEKi may include distinct metabolic adaptations, and then support the use of each inhibitor individually to fully characterize therapeutic vulnerabilities.

Once this switch and the crucial role of glutaminolysis in resistant cells were confirmed, we investigated the antitumor properties of glutaminolysis inhibition using the glutaminase inhibitor CB-839. Glutaminase inhibition reduced colony formation and induced apoptosis in these resistant cells in vitro and decreased tumor development in vivo. Moreover, rescue experiments with α-KG to replenish the TCA cycle confirmed that the antitumor effects of glutaminase inhibition are mediated by impaired TCA cycle fueling, as previously demonstrated in BRAF-mutant cells [7,40]. We also highlighted an enhanced effect of combining RTKi/MAPKi with glutaminase inhibition in slowing colony formation and inducing cell death. This combined effect was primarily associated with increased production of ROS, reduced glutathione levels, and inhibited conversion of glutamine to glutamate. This decreased glutathione levels and subsequent cell death were already demonstrated in lung cancer cells treated with glutaminase inhibitors [16]. Here, we further demonstrated that the antitumor effect of glutaminolysis inhibition differs between sensitive and resistant cells. Indeed, in resistant cells, cytotoxicity was mainly linked to decreased energy production, despite an effect on GSH and ROS levels in these cells, which were comparatively less sensitive to oxidative stress. While in sensitive cells, CB-839 cytotoxicity was primarily associated with reduced GSH production and increased oxidative stress.

Our study is the first to evaluate this therapeutic combination in BRAF and non-BRAF mutant melanoma lines, demonstrating a general mechanism of resistance through a metabolic switch. Moreover, we reported the mechanistic basis underlying the combined effect, including increased ROS levels, depleted glutathione, and suppressed glutamine-to-glutamate conversion. Supporting our findings, many studies reported treatment combinations with CB-839 in other cancers. In melanoma cell lines, CB-839 has been shown to synergize with PDK inhibitors to reduce proliferation by reprogramming cellular metabolism, unveiling potential metabolic co-vulnerabilities and suggesting a promising strategy for melanoma treatment [41]. In ovarian carcinoma and triple-negative breast cancer, glutaminolysis inhibition with CB-839 enhances sensitivity to mTOR inhibitors [42,43]. In esophageal squamous cell carcinoma, CB-839 significantly suppresses cell proliferation, colony formation, migration, invasion in vitro, and tumor growth in animals [44], and overcomes resistance to cyclin-dependent kinase inhibitors [45]. In colorectal cancer, CB-839 synergizes with the EGFR inhibitor cetuximab to decrease cell growth in vitro and tumor development in vivo [31]. Another recent study demonstrates that CB-839 enhances the anticancer effects of piperlongumine by inducing ferroptotic death in oral squamous cell carcinoma cells, suggesting a potential new strategy for cancer treatment [46]. Moreover, several studies have reported that targeting glutaminase is of particular interest in cancer treatment, as increased glutaminolysis has been associated with metastatic cells [47], and glutamate produced from glutamine is associated with cell invasion and migration [48,49,50].

Nowadays, phase I/II clinical trials have already determined the best doses for CB-839, demonstrating that it is generally well tolerated and produces robust inhibition of GLS in tumors [51,52]. The results are promising, with good responses reported in patients with advanced triple-negative breast cancer in combination with paclitaxel and in patients with relapsed/refractory leukemia who were heavily pretreated [51,53]. Moreover, another phase II study combining nivolumab with CB-839 showed partial responses in patients with melanoma, non-small cell lung cancer, and renal cell carcinoma that were progressing under immune checkpoint inhibitors alone (NCT02771626) (Appendix A). However, it is important to note that the clinical development of CB-839 was discontinued in 2023 due to limited efficacy across multiple tumor types. Although some patients exhibited responses, these were typically partial and short-lived, and no significant survival benefit was achieved in larger trials. To our knowledge, no current clinical trials with CB-839 are actively recruiting (Appendix A). This underlines a key limitation in translating glutaminase inhibition into a durable clinical benefit with this compound. For example, Li et al. explored the potent anticancer effect of the combination of CB-839 and 5-FU against mutant PIK3CA colorectal cancer tumors. Although no objective response in treated patients was reported, increased levels of neutrophil extracellular traps in posttreatment tumor biopsies (characterized by elevated cit-H3 levels) were associated with longer progression-free survival [54]. In another study in colorectal cancer, the combination of CB-839 and panitumumab was found to be safe and showed promising preliminary responses, but the study closed early due to CB-839 development termination (NCT03263429) [55]. Despite these potential clinical limitations, our study provides mechanistic insight into the metabolic vulnerabilities of resistant melanoma cells and highlights glutaminolysis as a relevant therapeutic target. Our findings thus support the development or repositioning of next-generation GLS inhibitors, particularly in rational combinations or in biomarker-selected patient populations. Further research is needed to overcome resistance mechanisms and identify strategies to enhance the clinical utility of glutaminase inhibition.

Once the metabolic switch was validated in cells, we examined the expression of enzymes/transporters involved in this change in patient samples. We showed that GLS overexpression correlated with poor survival in patients with metastatic melanoma. This clinical relevance reinforces the therapeutic potential of glutaminase inhibition. The prognostic value of GLS in melanoma was already indirectly observed in other studies. For example, a study on the long noncoding RNA *OIP5-AS1* found that *OIP5-AS1* increases *GLS* expression and is correlated with shorter survival in melanoma patients [56]. Another study on miR-137, which inhibits glutamine catabolism, also linked decreased miR-137 expression with shorter patient survival [57]. On the other hand, multiomics analyses highlighted the prognostic impact of GLS in different cancer types, as GLS overexpression was associated with shorter overall survival in breast, esophagus, head and neck, and blood cancers, while low expression of GLS correlated with longer overall survival in breast, colon, and esophagus cancers [58].

Interestingly, our study on biopsies from patients harboring ^V600E^BRAF mutations, further classified as responders or non-responders to vemurafenib, led to the determination of a metabolic gene signature that can predict the response to such targeted therapy. This signature needs to be validated in a larger cohort and could potentially be used to select patients most likely to benefit from BRAFi treatment. A similar glutamine metabolism scoring was proposed in hepatocellular carcinoma to predict prognosis and therapeutic resistance [59]. Although our metabolic gene signature was identified in a limited cohort of BRAF-mutant melanoma samples, it provides preliminary evidence for a link between glutamine metabolism and response to targeted therapy. Future work will aim to validate this signature in large publicly available melanoma datasets to assess its potential clinical utility for more personalized treatment strategies.

## 4. Materials and Methods

### 4.1. Inhibitors

The BRAF inhibitor dabrafenib, used in BRAF-mutated cells, and the MEK inhibitor pimasertib, used in NRAS-mutated cells, were obtained from Selleck Chemicals (Houston, TX, USA). The tyrosine kinase inhibitor dasatinib, used in cKIT-mutated cells, was obtained from Bristol-Myers Squibb (New York, NY, USA). The glutaminase inhibitor CB-839 (Telaglenastat^®^) is from MedChemExpress (Princeton, NJ, USA). Dimethyl α-ketoglutarate is from Sigma Aldrich (Saint Louis, MO, USA). Glutathione is from Amresco (Solon, OH, USA).

### 4.2. Cell Culture

The cell lines were derived from melanoma metastases by the Laboratory of Clinical and Experimental Oncology at the Institut Jules Bordet (Brussels, Belgium). We worked with three cell lines harboring different mutations and were either sensitive or had acquired resistance to targeted therapy: MM074 (^V600E^BRAF), MM161 (^Q61R^NRAS), and HBL (^D820Y^cKIT), and their corresponding cells with acquired resistance to the drugs (-R). BRAF, NRAS and cKIT mutations were assessed with the next-generation DNA sequencing for 48 genes from the cancer panel (TruSeq Amplicon-Cancer Panel, Illumina, San Diego, CA, USA) [60]. To develop these resistances, cells were chronically exposed to increasing concentrations of targeted therapies for 12 weeks (0.01 µM during Week 1 and 2; 0.05 µM during Week 3 and 4; 0.1 µM during Week 5 and 6; 0.5 µM during Week 7 and 8; 1 µM during Week 9 and 10; 2 µM during Week 11 and 12) [9,38]. Such resistances were previously validated showing significant increased IC50 (cristal violet staining) against these inhibitors. Cells were cultured in a Ham-F10 medium (Lonza, Basel, Switzerland) supplemented with 10% fetal bovine serum and 1% penicillin/streptomycin (both from Life Technologies, Carlsbad, CA, USA). Melanoma cells were monthly checked for mycoplasma contamination using a MycoAlert^®^ Mycoplasma Detection Kit (Lonza, Basel, Switzerland). Cell Line Authentication was performed using STR profiling with an AmpFLSTRTM IdentifilerTM PCR Amplification Kit (Thermo Fisher Scientific). DNA isolation was carried out from a cell pellet of 1 × 10^6^ cells, and 16 independent PCR systems were investigated and analyzed (Eurofins Genomics, Ebersberg, Germany).

### 4.3. Glutamine/Glutamate Dosage

The glutamine/glutamate ratio was evaluated by using the Glutamine/Glutamate-Glo^TM^ Assay (Promega, Madison, WI, USA) and following the kit instructions.

### 4.4. Clonogenic Assay

For the clonogenic assay, 2000 cells per well were plated in 6-well plates and cultured for 24 h. Cells were then treated with CB-839 for 2 weeks, fixed with 4% paraformaldehyde, and stained with a 4% crystal violet solution (Sigma Aldrich) for 30 min. The total number of colonies was counted under a microscope.

### 4.5. Cell Proliferation

Cell proliferation assays were performed in 96-well plates (Sarstedt, Antwerpen, Belgium). Cells were plated at 10,000 cells per well in 100 µL of a culture medium. The next day, cells were supplemented with 100 µL of a fresh medium containing CB-839 or with a vehicle for the control. After 72 h, cells were fixed with 1% glutaraldehyde (Sigma) for 15 min. Then the cells were stained with 4% crystal violet (Sigma) for 30 min. Cells were finally permeabilized with a solution of Triton X-100 (Sigma) for 90 min, and the absorbance was determined at 570 nm using a spectrophotometer (VersaMax-SoftMax Pro, Molecular Devices, San Jose, CA, USA). Results were normalized to the control (untreated cells).

### 4.6. RT-qPCR

Total RNAs were extracted from cell pellets using the InnuPrep RNA mini kit 2.0 (Westburg Life Sciences, Westburg, Germany). The concentration of total RNA was measured using a NanoDrop µlite (ThermoFisher Scientific, Waltham, MA, USA). All pairs of primers were analyzed for dissociation curves and melting temperatures. Reverse transcription and quantitative PCR were performed to quantify mRNA levels of *GAC*, *KGA*, *GLS2*, *GLUD1*, *SLC1A5*, *SLC7A5*, *SLC38A2*, *PPARGC1A*, *TFAM*, *PPRC1*, *NRF2*, *CAT*, *GSS*, *GSR1*, *GPX1*, and *18S* as a housekeeping gene (see Appendix A for primer sequences). Reverse transcription was performed using the Maxima First Strand cDNA Synthesis kit (ThermoFisher Scientific). The qPCR amplification was performed using the Takyon ROX SYBR 2X qPCR MasterMix dTTP blue (Eurogentec, Liège, Belgium). Relative mRNA expressions were calculated relative to untreated cells.

### 4.7. Polysome Profiling

To assess the polysome profile, 5,000,000 cells in flasks were treated with 100 µg/mL cycloheximide for 5 min at 37 °C and then harvested by scraping on ice in cold PBS containing 100 µg/mL cycloheximide. The cells were centrifuged at 400× *g* for 5 min at 4 °C and resuspended in 400 µL of a hypotonic buffer (5 mM Tris, pH 7.5, 1.5 mM KCl, 2.5 mM MgCl_2_, with complete protease and phosphatase inhibitors) containing 0.5% Triton X-100, 0.5% sodium deoxycholate, 2 mM DTT, 400 U/mL RNaseOUT, and 100 μg/mL cycloheximide. The lysates were loaded onto a 5–50% sucrose density gradient and centrifuged in an SW41 rotor at 38,000 rpm for 2 h at 4 °C. The polysome profiles were monitored, and the fractions were collected using a gradient fractionation system. The mRNA was finally extracted using LS-Trizol (ThermoFisher Scientific).

### 4.8. Flow Cytometry Assessment of Apoptosis, Oxidative Stress, Mitochondrial Levels, and ROS Level

Apoptosis and oxidative stress were assessed on the Muse^®^ Cell Analyzer (Merck, Brussels, Belgium) using the Muse Annexin V and Dead Cell Assay Kit and the Muse Oxidative Stress kit (Luminex, Austin, TX, USA), respectively. Mitochondrial content and ROS levels were also assessed by flow cytometry (Cyflow Space, Sysmex, Paris, France) using MitoTracker Green, MitoSOX Red, and H2DCFDA (ThermoFisher). Analyses were performed using FlowJo v10 software.

### 4.9. Immunofluorescence

Cells were fixed in 4% paraformaldehyde (Sigma Aldrich, Saint Louis, MO, USA) and blocked in PBS 1× (Klinipath, Olen, Belgium) containing 0.3% Triton X-100 and 5% normal goat serum (Cell Signaling Technology). The cells were incubated overnight at 4 °C with the anti-GLS antibody (rabbit, dilution 1:400) or anti-NRF2 antibody (rabbit, dilution 1:400) (Cell Signaling Technology) diluted in PBS 1×, 0.3% Triton X-100, and 1% BSA (Sigma). The secondary antibody used was Alexa FluorTM 488 Goat Anti-Rabbit IgG (Invitrogen, Renfrewshire, UK). Nuclei were stained using VectaShield-DAPI. Images were acquired using a confocal microscope (Nikon Ti2 A1RHD25, Tokyo, Japan).

### 4.10. Glutathione Measurement

Glutathione determination was performed using the Glutathione GSH-GSSG Assay Kit (Sigma) and following the manufacturer’s instructions.

### 4.11. Animal Study

Sixteen nude mice (Male CR ATH HO MOUSE, aged 28–34 days, Charles River, Saint-Germain-Nuelles, France) were used for the animal study, including eight control mice and eight mice treated with CB-839. A total of 5,000,000 MM074-R cells diluted in a 1:1 *v*/*v* saline/Cultrex Basement Membrane Extract, Type 3 (R&D Systems, Abingdon, UK), were injected subcutaneously into the right flank of the mice. From one week after injection, mice were treated with 160 mg/kg/day CB-839 by oral gavage. The mice weight, general behavior, and tumor size were monitored every 2 days. Tumor volume was calculated according to the formula (length × width × thickness)/2. After 3 weeks of treatment, the animals were euthanized by lethal intraperitoneal injection of 200 mg/mL Dolethal (pentobarbital sodium, Vetoquinol, Aartselaar, Belgium). At this stage, tumors did not exceed 1000 mm^3^. Tumors, hearts, lungs, livers, kidneys, and spleens were collected, weighed, fixed in Bouin solution, and embedded in paraffin for hematoxylin and eosin (HE) staining and histological analyses. The animal study involved sixteen nude mice injected with MM074-R cells to investigate tumor growth. Treatment with CB-839 was administered orally for three weeks, with regular monitoring of mice health and tumor size. Ethical approval was obtained from the University of Mons (SA-06-01), ensuring compliance with institutional guidelines for animal studies.

### 4.12. Patient Samples

A series of 84 skin and lymph node metastases were collected from patients with stage III or IV melanoma who underwent surgery at Institut Jules Bordet (Brussels, Belgium) between 1998 and 2009. Our study was performed in accordance with the REMARK guidelines [61,62]. The clinical characteristics of the patients are outlined in Table 1A. Biopsies from 8 patients with metastatic melanoma harboring ^V600E^BRAF mutations were collected and snap-frozen in liquid nitrogen before the initiation of first-line therapy with vemurafenib. Four patients were responders (partial or complete response as the best response using Response Evaluation Criteria In Solid Tumors (RECIST) 1.1 [63] or PET Response Criteria in Solid Tumors (PERCIST) [64]), and 4 were non-responders (progressive disease as the best response). The clinical characteristics of patients are presented in Table 1B. Patient samples from melanoma cases at Institut Jules Bordet were collected ethically, following approval from the institute ethics committee (CE2023) and adherence to REMARK guidelines.

### 4.13. Immunohistochemistry

IHC using the rabbit antibody raised against GLS1 (ThermoFisher Scientific) was performed on a BenchMark XT System (Ventana, Tucson, AZ, USA). The detection of the primary antibody was performed using the ultraView Universal Alkaline Phosphatase Red Detection Kit (Ventana, Tucson, AZ, USA). Immunostaining was evaluated under a microscope, and a score from 0 to 300 was calculated by adding the percentage of tumor cells with no (=0), weak (=1), intermediate (=2), or strong (=3) staining intensity (Figure 6A).

### 4.14. Statistical Analysis

Statistical analyses were performed using IBM SPSS Statistics 21 software (IBM, Ehningen, Germany). Data were compared using the Mann–Whitney test (non-parametric) or ANOVA followed by Tukey post hoc test (parametric), depending on the normality of the distribution. A *p*-value ≤ 0.05 was considered statistically significant (* *p* ≤ 0.05; ** *p* ≤ 0.01; *** *p* ≤ 0.001; NS = Not Significant). For FACS data, statistical analyses were performed using ANOVA followed by Tukey post-hoc tests to compare peak maxima. For the quantification and corresponding statistical analyses of immunofluorescence images, green fluorescence (pixel count) was measured and compared. Animal weight progression was monitored for 3 weeks following the start of treatment and statistically analyzed using a linear model for repeated measures, with additional Mann–Whitney tests performed on days 23 and 27. For all experiments, a minimum of 3 biological replicates were performed. For the survival curve study, overall survival (OS) analyses were conducted by calculating Kaplan–Meier curves, while Cox regression models were used to calculate the hazard ratio (HR) and significance.

## 5. Conclusions

To conclude, our study demonstrates that melanoma cells with different mutational profiles undergo a similar metabolic switch when they develop resistance to RTKi or MAPKi treatment, and that this switch relies on glutaminolysis. In this context, glutaminolysis inhibition by CB-839 is a promising treatment option, as it enhanced the antitumor effects of RTKi/MAPKi in sensitive cells and was also effective in killing cells resistant to RTKi/MAPKi. Of note, CB-839 also appears to be of particular interest in combination with other treatment strategies, as it is already being evaluated in clinical trials across various cancer types, including melanoma resistant to immunotherapy [37]. To our knowledge, our study is the first to highlight the antitumor effect of CB-839 in melanoma cells with acquired resistance to targeted therapies, supporting further clinical investigation of glutaminolysis inhibition in this context. Finally, we suggest that assessing the expression of specific metabolism-associated genes in patient biopsies prior to BRAFi treatment could serve as a useful signature to predict responsiveness to targeted therapy against BRAF. This approach may help identify patients who are more likely to benefit from such treatments.

## Figures and Tables

**Figure 1 ijms-26-08241-f001:**
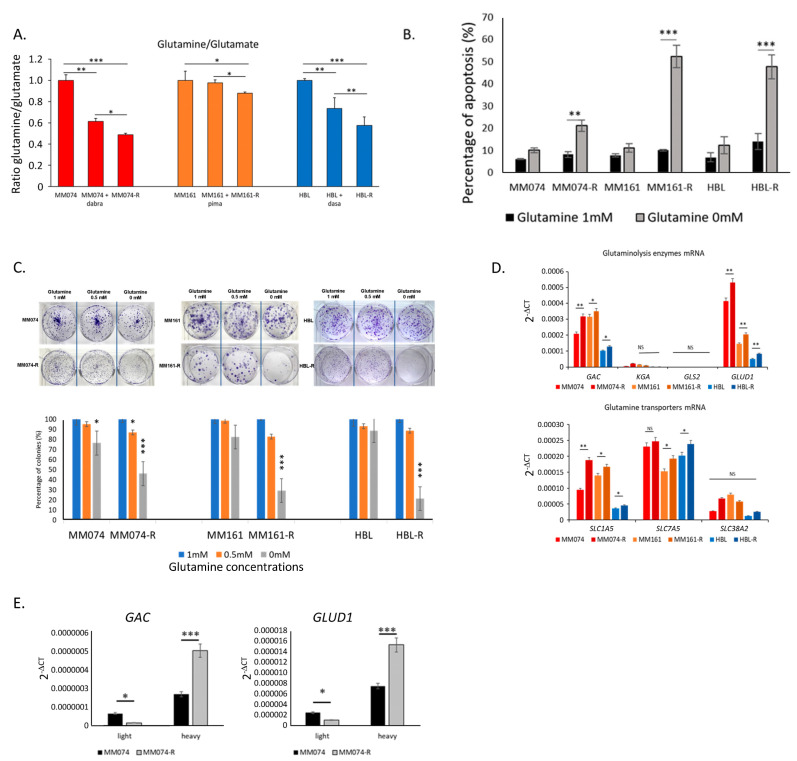
Differential involvement of glutaminolysis in sensitive and resistant melanoma cells to RTKi/MAPKi. (**A**) The glutamine/glutamate ratio assessed after glutamine and glutamate quantification in the intracellular medium of sensitive and resistant cells exposed to RTKi/MAPKi (1 µM) for 24 h. (**B**) Impact of glutamine deprivation for 48 h on apoptosis induction (% of apoptotic cells). (**C**) The clonogenic assay evaluating the influence of glutamine deprivation (0,0.5 vs. 1 mM) in sensitive and resistant cells after 2 weeks of culture. (**D**) mRNA levels of glutaminolysis enzymes (*GAC*, *KGA*, *GLS2*, *and GLUD1*) and transporters (*SLC1A5*, *SLC7A5*, *and SLC38A2*) determined through qPCR analyses in sensitive and resistant cell lines. (**E**) mRNA expression of the glutaminolysis enzymes *GAC* and *GLUD1* in light and heavy fractions obtained via polysome profiling. Statistical significance, *: *p* < 0.05, **: *p* < 0.01, ***: *p* < 0.001.

**Figure 2 ijms-26-08241-f002:**
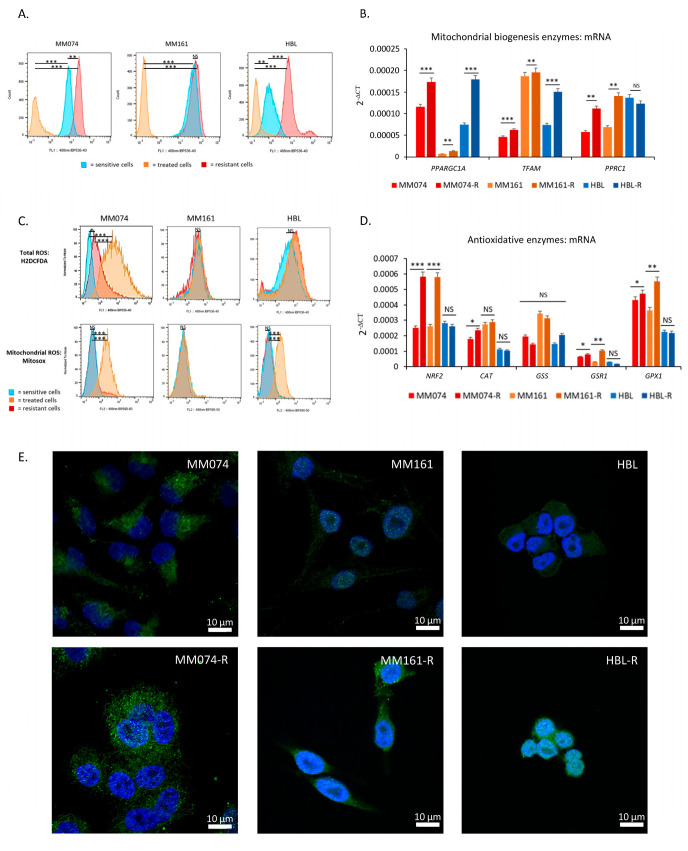
Mitochondrial and total ROS content in sensitive and resistant melanoma cells to RTKi/MAPKi. (**A**) Mitochondrial content evaluated by flow cytometry using MitoTracker Green. (**B**) mRNA levels of factors involved in mitochondrial biogenesis determined by qPCR analyses. (**C**) Levels of total and mitochondrial ROS content assessed by flow cytometry using H2DCF-DA (total ROS) and Mitosox (mitochondrial ROS). (**D**) mRNA levels of a transcription factor (*NRF2*) and enzymes (*CAT*, *GSS*, *GSR1*, and *GPX1*) involved in ROS detoxification determined by qPCR analyses. (**E**) Immunofluorescence image of NRF2 (green) captured by confocal microscopy for the 6 lines (scale bars = 10 µm), DAPI staining (blue) for the nucleus. (quantification of green fluorescence is presented in Appendix A). Statistical significance, *: *p* < 0.05, **: *p* < 0.01, ***: *p* < 0.001, NS: non significant.

**Figure 3 ijms-26-08241-f003:**
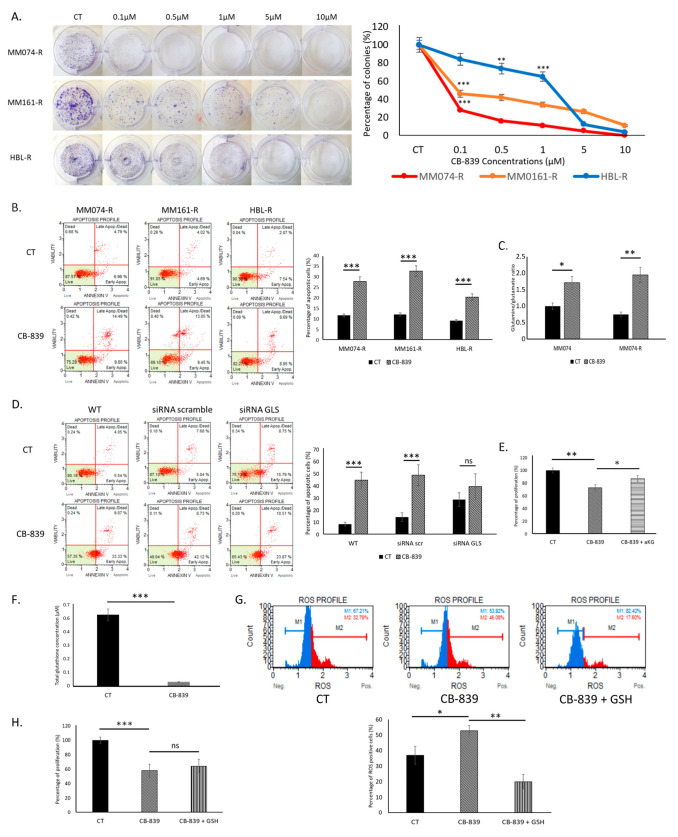
Effect of the glutaminase inhibitor CB-839 in melanoma cells with resistance to RTKi/MAPKi. (**A**) A clonogenic assay assessing the impact of CB-839 (0.1–10 µM) after two weeks of exposure. (**B**) Effect of CB-839 (10 µM) for 48 h on apoptosis induction. (**C**) Effect of CB-839 (10 µM) for 24 h on the intracellular glutamine/glutamate ratio in BRAF-mutated MM074-R cells. (**D**) Impact of GLS knock-down on apoptosis induction by CB-839 (10 µM) for 48 h in BRAF-mutated MM074-R cells. (**E**) Effect of α-KG supplementation (5 mM) on the antiproliferative effect (crystal violet staining) induced by CB-839 (10 µM) in BRAF-mutated MM074-R cells for 72 h. (**F**) Evaluation of the total glutathione (GSH) level in cells exposed to CB-839 (10 µM) for 24 h in BRAF-mutated MM074-R cells. (**G**) A flow cytometry assay evaluating the effect of CB-839 (50 µM) alone or combined with GSH (5 mM) for 2 h on ROS induction in BRAF-mutated MM074-R cells (upper panel: FACS plots; lower panel: FACS quantification). (**H**) Effect of GSH supplementation (0.1 M) on the antiproliferative effect induced by CB-839 (10 µM) for 72 h in BRAF-mutated MM074-R cells. Statistical significance, *: *p* < 0.05, **: *p* < 0.01, ***: *p* < 0.001.

**Figure 4 ijms-26-08241-f004:**
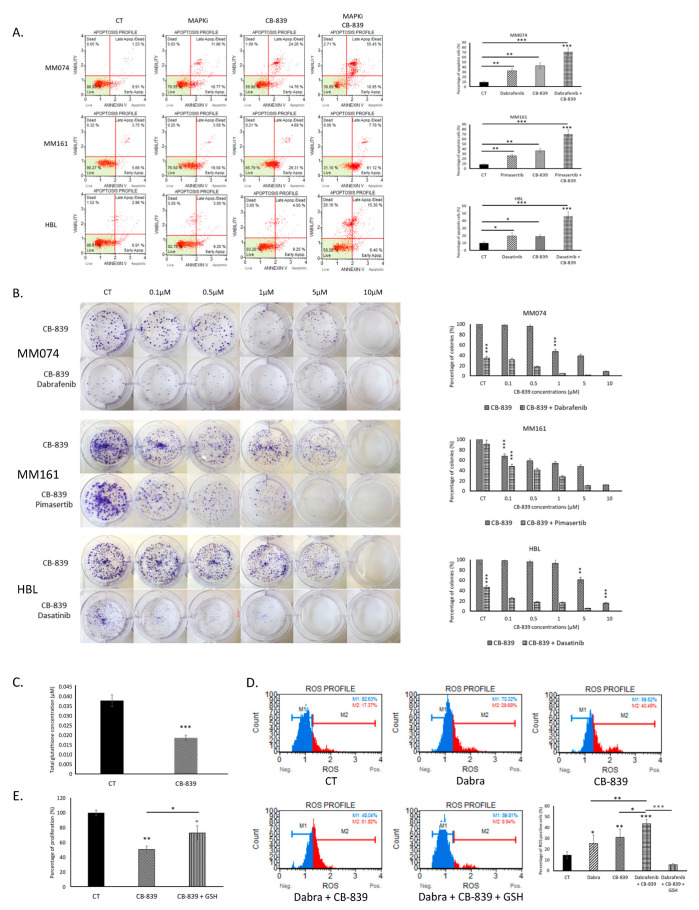
Effect of the glutaminase inhibitor CB-839 on sensitive melanoma cells to RTKi/MAPKi. (**A**) Impact of CB-839 (10 µM) for 48 h, alone or in combination with RTKi/MAPKi (1µM), on apoptosis induction. (**B**) A clonogenic assay evaluating the effect of CB-839 (0.1–10 µM) alone or in combination with RTKi/MAPKi (1 µM) after two weeks of exposure to the drug(s). (**C**) Assessment of the total glutathione (GSH) level in dabrafenib-sensitive BRAF-mutated MM074 cells exposed to CB-839 (20 µM) for 24 h. (**D**) A flow cytometry assay evaluating the effect of CB-839 (50 µM) alone or in combination with dabrafenib (1 µM), with or without GSH supplementation (0.1 M) for 2 h, on ROS induction in the sensitive BRAF-mutated MM074 cells. (**E**) Effect of GSH supplementation (0.1 M) on the antiproliferative effect (crystal violet staining) induced by CB-839 (10 µM) for 72 h in sensitive BRAF-mutated MM074 cells. Statistical significance, *: *p* < 0.05, **: *p* < 0.01, ***: *p* < 0.001.

**Figure 5 ijms-26-08241-f005:**
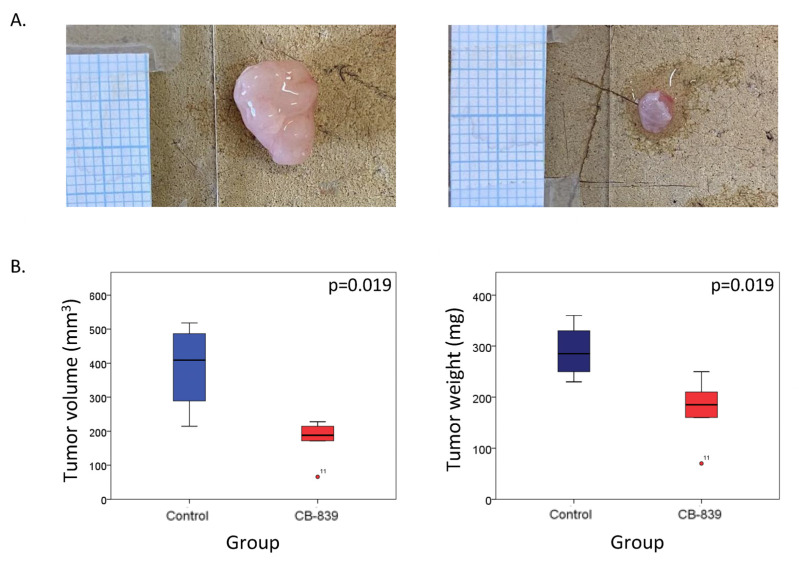
Effect of CB-839 in nude mice xenografted with MM074-R cells in the right flanks. (**A**) Representative images of an MM074-R tumor excised from a control animal (left photo) compared to a tumor excised from a mouse treated with CB-839 (160 mg/kg daily) for three weeks (right photo). Millimeter paper allows for size comparison. (**B**) Box plots and the Mann–Whitney test illustrating tumor volume and tumor weight assessed after euthanasia of mice in both control and CB-839 treated groups. Point 11 is considered an outlier because it falls outside the whiskers of the box plot.

**Figure 6 ijms-26-08241-f006:**
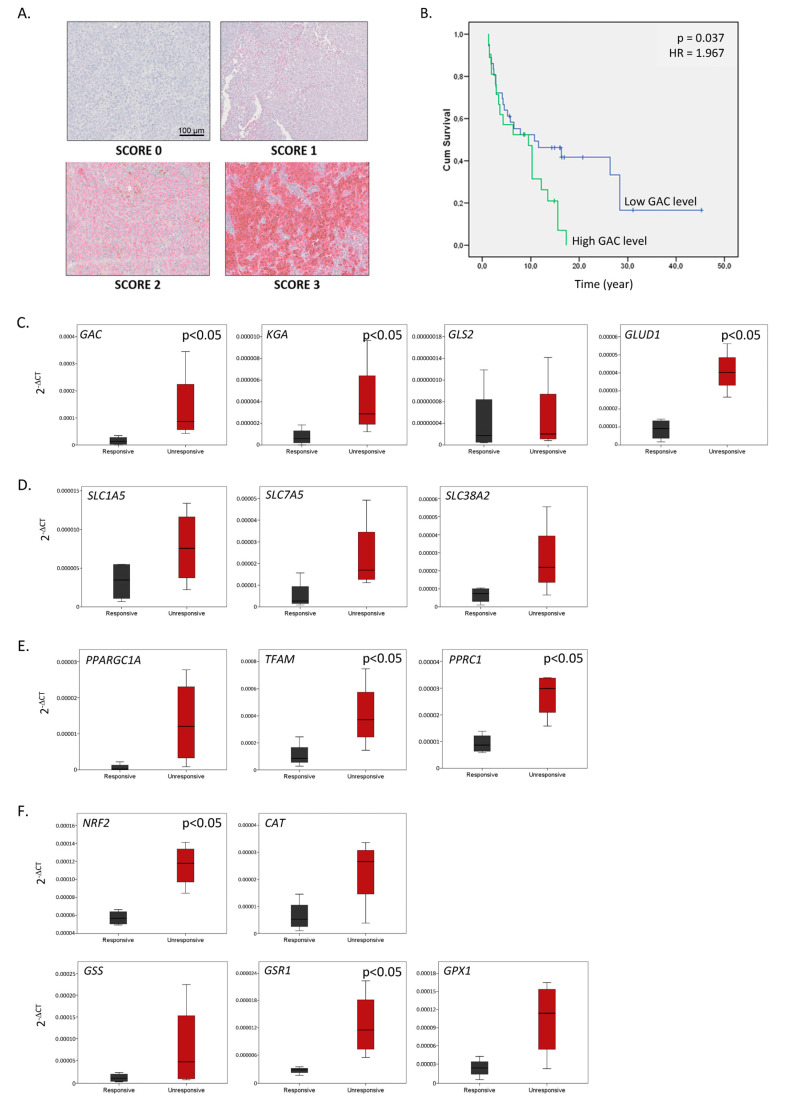
Glutaminolysis enzyme and transporter expression in patient tumors. (**A**) Representative immunohistochemistry (IHC) of GAC in melanoma metastases. Examples of staining intensity scored at 0 (none), 1 (low), 2 (intermediate), and 3 (high). Magnification 40×. (**B**) Kaplan–Meier curves comparing overall survival between groups with low (blue) and high (green) GAC protein levels; “+” are censored data (patients alive or lost to follow-up). (**C**) mRNA levels of glutaminolysis enzymes, (**D**) mRNA levels of glutamine transporters, (**E**) mRNA levels of mitochondrial biogenesis-linked enzymes, and (**F**) mRNA levels of factors involved in antioxidative defenses. (**C**–**F**) Evaluated by qPCR in 8 patient biopsies collected prior to targeted therapies and categorized as responders (black, n = 4) or non-responders (red, n = 4) to the BRAF inhibitor vemurafenib.

**Table 1 ijms-26-08241-t001:** Characteristics of melanoma samples and patients. (**A**) Tumor samples used to evaluate GAC protein expression by IHC (n = 84). (**B**) All tumors express V600E BRAF, and all patients were treated with vemurafenib after sample collection. NM: nodular melanoma. SSM: superficial spreading melanoma. LN: lymph node metastasis. SK: skin metastasis. OS: overall survival.

A.						
Parameters				N	Median	Range
Metastatic site (lymph nodes/skin)				48/36		
Gender (female/male)				50/34		
Age at diagnosis of primary melanoma (years)				84	55	20–77
Stage at sample collection (III/IV)				54/30		
Breslow thickness (<1/≥1) (mm)				19/65	2.6	0.3–28.0
Ulceration of primary lesion (no/yes)				29/55		
Invaded lymph nodes at primary (0/≥1)				40/23		
Overall survival (years)				80	4.9	0.8–45.9
Status (alive/dead)				26/58		
B.						
Sample Number	Melanoma Type	Metastatic Site	Response to Vemurafenib	Duration of Vemurafenib Treatment (Months)	OS (Months)	OS Status
#1	NM	LN	Responder	4	65	Dead
#2	SSM	LN	Responder	9	70	Dead
#3	SSM	LN	Responder	39	101	Dead
#4	SSM	LN	Responder	89	92	Alive
#5	SSM	LN	Non-responder	4	23	Dead
#6	NM	SK	Non-responder	3	79	Dead
#7	SSM	SK	Non-responder	4	28	Dead
#8	NM	LN	Non-responder	3	30	Dead

## Data Availability

The data presented in this study are available on request from the corresponding author. The data are not publicly available due to confidentiality restrictions.

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
