# Peer review of "Therapeutic Potential of Glutaminase Inhibition Targeting Metabolic Adaptations in Resistant Melanomas to Targeted Therapy"

_ijms, 2025, doi:10.3390/ijms26178241_

Round 1
Reviewer 1 Report
Comments and Suggestions for Authors
Dear Authors,
This is a well-executed and timely study exploring glutaminase inhibition with CB-839 as a strategy to overcome therapy resistance in melanoma. The use of multiple melanoma genotypes, in vitro and in vivo models, and patient-derived data strengthens the findings. To improve clarity and reproducibility, please:
- Clearly state the central hypothesis at the end of the introduction.
- Provide details on how resistant cell lines were generated and validated.
- Clarify sample sizes and treatment conditions in the in vivo experiments.
- Adjust wording around “synergistic” effects unless formal synergy analysis was conducted.
- Rephrase or temper claims about “enhanced immunotherapy response,” as no immunotherapy was directly tested in this study.
- Briefly report and reference key supplementary results (e.g., α-KG rescue).
- Streamline the language in a few dense sections and check for minor inconsistencies or formatting issues.
These changes will improve transparency and alignment between data and interpretation.
Comments on the Quality of English LanguageThe quality of English in the manuscript is generally good, with clear scientific language and logical flow. The authors successfully convey complex experimental results and interpretations with precision. However, several sentences—particularly in the Results and Discussion sections—are dense or overly technical, and could benefit from slight rephrasing for improved readability. Minor grammatical inconsistencies and occasional awkward phrasing are present but do not impede understanding. A light language edit or professional proofreading is recommended to enhance clarity, streamline complex sentences, and correct small stylistic issues.
Author Response
Reviewer 1
Dear Authors,
This is a well-executed and timely study exploring glutaminase inhibition with CB-839 as a strategy to overcome therapy resistance in melanoma. The use of multiple melanoma genotypes, in vitro and in vivo models, and patient-derived data strengthens the findings. To improve clarity and reproducibility, please:
- Clearly state the central hypothesis at the end of the introduction. We better explain the hypothesis on lines 78-81 : we hypothesize that melanoma cells resistant to RTKi/MAPKi undergo a metabolic switch toward glutaminolysis, and that this dependency can be therapeutically targeted using the glutaminase inhibitor CB-839 to overcome resistance and restore antitumor efficacy.
- Provide details on how resistant cell lines were generated and validated. We added details on lines 442-446 : MM074 (V600EBRAF), MM161 (Q61RNRAS) and HBL (D820YcKIT) and their corresponding cells with acquired resistance to the drugs (-R). Resistant cells were developed by chronic treatment with increasing concentrations of the inhibitors against mutated targets and validated showing increased IC50 (cristal violet staining, data not shown) against these inhibitors.
- Clarify sample sizes and treatment conditions in the in vivo experiments. We added details on lines 516-517 : Sixteen nude mice (Male CR ATH HO MOUSE, aged 28–34 days, Charles River, Saint-Germain-Nuelles, France) were used for the animal study, including eight control mice and eight mice treated with CB-839.
- Adjust wording around “synergistic” effects unless formal synergy analysis was conducted. We adjusted the sentences on lines 218-220 : We observed an increased effect of the combination on apoptosis induction (Figure 4A) and colony formation (Figure 4B) compared to either treatment alone, suggesting an enhanced efficacy when the two drugs are used together.
On lines 344-348 : We also highlighted an enhanced effect of combining RTKi/MAPKi with glutaminase inhibition in slowing colony formation and inducing cell death. This combined effect was primarily associated with increased production of ROS, reduced glutathione levels, and inhibited conversion of glutamine to glutamate.
On line 358-359 : Moreover, we reported the mechanistic basis underlying the combined effect, including increased ROS levels, depleted glutathione, and suppressed glutamine-to-glutamate conversion. And on line 567-568 : In this context, glutaminolysis inhibition by CB-839 is a promising treatment option, as it enhanced the antitumor effects of RTKi/MAPKi in sensitive cells and was also effective in killing cells resistant to RTKi/MAPKi.
- Rephrase or temper claims about “enhanced immunotherapy response,” as no immunotherapy was directly tested in this study. We rephrased the sentence at Lines 569-571: Of note, CB-839 also appears to be of particular interest in combination with other treatment strategies, as it is already being evaluated in clinical trials across various cancer types, including melanoma resistant to immunotherapy (43)
- Briefly report and reference key supplementary results (e.g., α-KG rescue). Additional figures and tables were already referenced in the text (in red); their legends have now been expanded with more details.
- Streamline the language in a few dense sections and check for minor inconsistencies or formatting issues. Thank you for your feedback. The Results and Discussion sections have been revised to improve clarity.
These changes will improve transparency and alignment between data and interpretation.
The quality of English in the manuscript is generally good, with clear scientific language and logical flow. The authors successfully convey complex experimental results and interpretations with precision. However, several sentences—particularly in the Results and Discussion sections—are dense or overly technical and could benefit from slight rephrasing for improved readability. Minor grammatical inconsistencies and occasional awkward phrasing are present but do not impede understanding. A light language edit or professional proofreading is recommended to enhance clarity, streamline complex sentences, and correct small stylistic issues.
Reviewer 2 Report
Comments and Suggestions for Authors
Review of Therapeutic potential of glutaminase inhibition targeting metabolic adaptations in resistant melanomas to targeted therapy
The authors claim that glutaminase inhibition is beneficiary in melanoma resistant to targeted therapy. They identify glutaminase dependency being elevated in single-agent resistant melanoma tumors which in turn can be targeted with a glutaminase inhibitor GB-839.
General comments
In general, the figures need to be in higher resolution.
The cell lines and especially the generation of resistant cell lines should be described either in the result or the method sections. How were the cell lines made resistant?
Specific comments:
Figure 1A: Please add y-axis label,
Figure 1B: please add bar graph showing apoptosis levels with statistics
Figure 2A: C please add a statistical evaluation of the FACS results.
Figure 2E: Please add bar graph showing the NRF2 levels with statistics
Line 142: There is no statistical analysis for ROS levels in Fig. 2C.
Figure 3a: A line graph would be more appropriate for this kind of colony formation assay.
Figure 3B,D: The bar graphs data do not fit to the FACS results, Please recalculate. Were late and early apoptosis added? I would use the early apoptosis as only this has a significant increase
Line 185: What is the reasoning using the MM074 line? Please explain.
Line 208 and discussion: Is the CB-839 - RTKi/MAPKi combination treatment additive or synergistic? Please have a statistical evaluation to show that.
Figure 4 is identical to figure 3!
Line 237: Please show that in the supplement.
Figure 6A: higher resolution please. Add size scale.
Figure 6C-F: please label Y-axis as delta CT
Table 1: Please add a supplemental data table with patient Braf status and other characteristics.
Author Response
Reviewer 2:
The authors claim that glutaminase inhibition is beneficiary in melanoma resistant to targeted therapy. They identify glutaminase dependency being elevated in single-agent resistant melanoma tumors which in turn can be targeted with a glutaminase inhibitor CB-839.
General comments
In general, the figures need to be in higher resolution.
The cell lines and especially the generation of resistant cell lines should be described either in the result or the method sections. How were the cell lines made resistant? We added details on lines 442-446 : MM074 (V600EBRAF), MM161 (Q61RNRAS) and HBL (D820YcKIT) and their corresponding cells with acquired resistance to the drugs (-R). Resistant cells were developed by chronic treatment with increasing concentrations of the inhibitors against mutated targets and validated showing increased IC50 (cristal violet staining, data not shown) against these inhibitors.
Specific comments:
Figure 1A: Please add y-axis label. We added “ratio glutamine/glutamate” as y-label (line 96)
Figure 1B: please add bar graph showing apoptosis levels with statistics. We added the bar graph (line 96)
Figure 2A: C please add a statistical evaluation of the FACS results. We have now completed the description of this analysis: “Our data showed that MM074-R and HBL-R resistant cells exhibited higher mitochondrial content compared to their sensitive counterparts (Figure 2A). No change was observed in the MM161 line. Notably, treatment of sensitive cells with RTKi/MAPKi significantly decreased their mitochondrial levels”.
Figure 2E: Please add bar graph showing the NRF2 levels with statistics. We did not quantify the NRF2 levels as we just wanted to see its localization.
Line 142: There is no statistical analysis for ROS levels in Fig. 2C.
Figure 3a: A line graph would be more appropriate for this kind of colony formation assay. We change the type of graph with a line graph.
Figure 3B,D: The bar graphs data do not fit to the FACS results, Please recalculate. Were late and early apoptosis added? I would use the early apoptosis as only this has a significant increase. The FACS represents a single replicate and the graph shows three replicates. The graphs show total apoptosis. Early and late apoptosis are both represented in FACS.
Line 185: What is the reasoning using the MM074 line? Please explain. The focus on these two cell lines is now explained in lines 192–197 : Because the MM074-R cell line was the most sensitive to CB-839 (Figure 3A) and is clinically relevant in the context of BRAF inhibitor therapies, we specifically focus on both MM074 and MM074-R cells to validate whether CB-839 reduces cell survival by inhibiting glutaminolysis. To do so, we quantified glutamine and glutamate levels in MM074 and MM074-R cells and observed an increased glutamine/glutamate ratio in cells treated with CB-839, indicating a decrease in the conversion of glutamine into glutamate (Figure 3C).
Line 208 and discussion: Is the CB-839 - RTKi/MAPKi combination treatment additive or synergistic? Please have a statistical evaluation to show that. Although a synergistic effect appears to be observed with the combination, we did not assess the degree of additivity or synergy using the Chou–Talalay method or CalcuSyn software. Therefore, we have now adjusted the wording related to “synergistic” effects throughout the text, as also advice by the Reviewer 1.
Figure 4 is identical to figure 3! Apologies for the upload error. Please find the correct Figure 4 presented now.
Line 237: Please show that in the supplement. Please find additional statistical analysis regarding animal weight between untreated and treated mice, and showing no significant change for mice xenografted with MM074 and animals xenografted with MM074-R, line 250-254.
Figure 6A: higher resolution please. Add size scale. Please find Figure 6A with the highest resolution we have and size scale.
Figure 6C-F: please label Y-axis as delta CT. The Y-label axis has been changed
Table 1: Please add a supplemental data table with patient Braf status and other characteristics. Please find the new Tables 1A and 1B, which include additional clinical data on the patient tissues used.
Round 2
Reviewer 2 Report
Comments and Suggestions for Authors
Please see attached PDF document

Author Response
Reviewer 2:
Please see my comments in blue
General comments
The cell lines and especially the generation of resistant cell lines should be described either in the result or the method sections. How were the cell lines made resistant? We added details on lines 442-446 : MM074 (V600EBRAF), MM161 (Q61RNRAS) and HBL (D820YcKIT) and their corresponding cells with acquired resistance to the drugs (-R). Resistant cells were developed by chronic treatment with increasing concentrations of the inhibitors against mutated targets and validated showing increased IC50 (cristal violet staining, data not shown) against these inhibitors.
How were the mutations introduced? Please add references for that. Also, add some more details on the resistant cell line generation. It would be of great value for this study doing RNAseq on the sensitive parental cell line vs the resistant ones. Are they available from a repository? If yes, please add that information.
Additional information and references are added in lines 445-451: BRAF, NRAS and cKIT mutations were assessed with the next-generation DNA sequencing for 48 genes from the cancer panel (TruSeq Amplicon-Cancer Panel, Illumina, San Diego, CA, USA) (Sabbat Frontiers in Medicine 2023). To develop these resistances, cells were chronically exposed to increasing concentrations of targeted therapies for 12 weeks (0.01 µM during Week 1 and 2; 0.05 µM during Week 3 and 4; 0.1 µM during Week 5 and 6; 0.5 µM during Week 7 and 8; 1 µM during Week 9 and 10; 2 µM during Week 11 and 12) (Soumoy Cancers 2020, Soumoy Cancer Cell int 2024).
Specific comments:
Figure 2A: C please add a statistical evaluation of the FACS results. We have now completed the description of this analysis: “Our data showed that MM074-R and HBL-R resistant cells exhibited higher mitochondrial content compared to their sensitive counterparts (Figure 2A). No change was observed in the MM161 line. Notably, treatment of sensitive cells with RTKi/MAPKi significantly decreased their mitochondrial levels”.
There is still no statistics in Fig.2AC to claim significance. Please add a statistical evaluation!
As required, we have performed statistical analyses of the FACS data using ANOVA and Tukey post-hoc tests, comparing the peak maxima from 3 replicate experiments and text have been adapted to present the significant data in lines 131-137 and 154-162.
Figure 2E: Please add bar graph showing the NRF2 levels with statistics. We did not quantify the NRF2 levels as we just wanted to see its localization.
But quantification of repeated experiments would support your conclusion, otherwise this is just speculation on a single experiment.
Quantification from 3 independent experiments and the corresponding statistical analyses are now presented in Additional Figure 3 to complement Figure 2E and in lines 169-171.
Line 142: There is no statistical analysis for ROS levels in Fig. 2C.
See comments above.
Please see the responses above.
Line 237: Please show that in the supplement. Please find additional statistical analysis regarding animal weight between untreated and treated mice, and showing no significant change for mice xenografted with MM074-R and treated or not with CB-839, line 250-254.
Where is that data?
We apologize for the oversight; the data can be found in Additional Figure 4.
Figure 6A: higher resolution please. Add size scale. Please find Figure 6A with the highest resolution we have and size scale.
There is no way of identifying membranes-specific CAG protein with this resolution. Please remove the cell membrane statement in line 267.
It has been corrected in lines 269-270.

Round 3
Reviewer 2 Report
Comments and Suggestions for Authors
All comments have been answered to my satisfaction.